# SUBWORD EMBEDDING FROM BYTES AGAINST EMBEDDING-BASED ATTACKS

## ABSTRACT

NLP models have grown as a powerful technology and impact our social life like never before, along with rising concerns in practical applications including privacy invasion and high computational cost. While federated learning alleviates these problems, attackers can still recover the private training data of victim clients by leveraging the transmitted model parameters and gradients. Protecting against such attacks of private information leakage remains an open challenge. We propose Subword Embedding from Bytes (SEB) as a novel solution that can protect privacy while maintaining efficiency and accuracy. Our experiments demonstrate that SEB can effectively protect against embedding-based attacks, which recover the sentences in a batch of text data, based on the gradients in federated learning. As a defense, SEB does not compromise the model's accuracy. We also verify that SEB obtains comparable and even better results over traditional subword embedding methods in machine translation, sentiment analysis, and language modeling.

## 1 INTRODUCTION

Natural Language Processing (NLP), such as Large Language Models (LLMs) and Machine Translations (MT), has made noticeable advancements in performance over the last decades, which is partially attributed to the availability of larger datasets and richer computational resources. Since most data are from users, their privacy concerns play an increasingly critical role. Federated learning (FL) offers a promising approach for preserving user data privacy, enabling training shared models across multiple clients without transferring the data to a central server. However, data leakage attack is a severe problem in federated learning. Although only the model updates are sent to the central server, sensitive information can be leaked through the model updates so that adversaries can use them to reconstruct the original data, compromising the user's privacy. Figure 1(a) demonstrates an FL framework, and Figure 1(b) shows how embedding-based attacks work as in Gupta et al. (2022). In the illustrated example, the attacker extracts all candidate words in a batch of data from the embedding gradients and can easily reconstruct the text with beam search and reordering.

The reason why the original text can be easily reconstructed is that most NLP models are typically based on word/subword tokenization such as Byte Pair Encoding (BPE) (Sennrich et al., 2015; Kudo & Richardson, 2018). There is a one-to-one mapping between a word/subword and an embedding vector. When a vector is updated, we can directly look up the corresponding word/subword. Therefore, we aim to design a one-to-many mapping between words/subwords and embedding vectors to increase the difficulty of the simple lookup. An intuitive idea is applying the byte embedding method because the same bytes are repeatedly used for multiple subwords. Thus, retrieving input subwords with the updated byte embeddings is harder, which makes the byte embedding in NLP models a potential defense. As shown in Figure 1(c), although the attacker extracts a bag of bytes, the candidate subword number is much larger than using subword embeddings. Therefore, the search space is the whole vocabulary and the recovery is more random.

Although previous work shows byte-based models can reduce vocabulary size compared to the subword models Xue et al. (2022); Shaham & Levy (2021); Zhang & Xu (2022), directly applying existing byte encodings to enhance privacy and efficiency faces two major challenges: First, the lack of explicit word boundaries makes it difficult to disambiguate between words and similar byte sequences. Smaller textual granularity cannot show the semantic meaning of each word leading to a

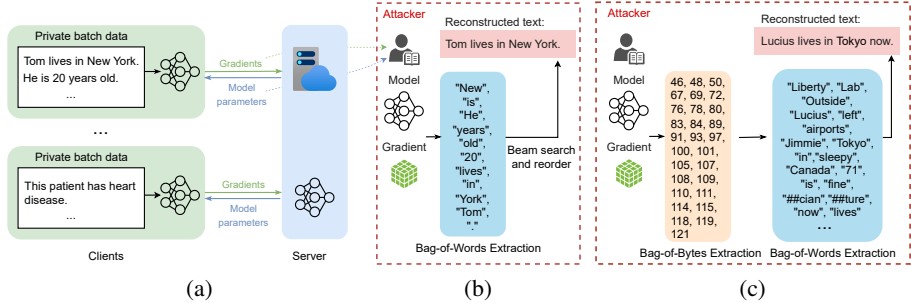

Figure 1: An attack example of recovering text in FL. (a): An FL framework. (b) and (c): Recovering text using embedding gradients of subwords and bytes.

less interpretable and analyzable model. Second, byte-based models tend to be more computationally expensive, as input sequences become much longer after byte tokenization.

To address these challenges in byte-based models, we propose to encode subwords with bytes and aggregate the byte embeddings to obtain a single subword embedding. This approach allows us to preserve the advantages of a small vocabulary and maintain the original subword boundaries with the same sequence length. The procedure consists of three steps: (1) Construct a mapping between subwords and bytes. (2) Convert the input text into a byte sequence. (3) Retrieve the corresponding byte embeddings and aggregate them back into subword embeddings using a feed-forward network, while ensuring that the subword boundaries are maintained. By adopting this approach, we can leverage the privacy protection provided by bytes while preserving the semantic meaning of the input sequence without increasing its length.

**Our main contributions are**:

- We introduce a novel and universal text representation method SEB, which achieves a constant vocabulary size (e.g., 256) without increasing the input sequence length.

- We verify that the proposed SEB can protect NLP models against data leaking attacks on privacy-sensitive information from the gradients of embeddings in federated learning. To the best of our knowledge, our work is the first one to study privacy preservation with byte representations in federated learning.

- We demonstrate that SEB achieves comparable and even superior performance in various NLP tasks compared to subword baselines with less space complexity and better privacy preservation.

## 2 RELATED WORK

**Attacks and defenses in language model**    Contrary to the belief that gradient sharing is safe in federated learning, many works show that it is possible to infer private information about the training data from the shared gradients (Zhu et al., 2019; Zhu & Blaschko, 2021; Deng et al., 2021; Balunovic et al., 2022; Gupta et al., 2022). Some recent works consider the reconstruction as an optimization task (Zhu et al., 2019; Deng et al., 2021; Balunovic et al., 2022). The attacker updates its dummy inputs and labels to minimize the distance between the gradients that the victim uploaded and the gradients the attacker calculated based on its dummy inputs and labels. Gupta et al. (2022) shows that the attackers can reconstruct a set of words with the embedding gradients, then apply beam search and reorder with a pretrained language model for input recovery.

One potential strategy to mitigate the gradient inversion attack described in Zhu et al. (2019); Deng et al. (2021); Balunovic et al. (2022) is to encrypt the communicated gradients or make them not directly inferable. However, encryption of the gradients requires special setups and could be costly to implement. Moreover, it does not provide effective protection against server-side privacy leakage (Aono et al., 2017; Huang et al., 2021; Fang & Qian, 2021). Differential privacy is another approach to protect against privacy attacks, but it may have an impact on model performance (Zhu et al., 2019; Wei et al., 2020; Yin et al., 2021; Li et al., 2021). While Zhang & Wang (2021)

proposed a secure federated learning framework that can prevent privacy leakage based on gradient reconstruction, it does not effectively address the retrieval of a bag of words from the embedding matrix gradients, as proposed in Gupta et al. (2022).

**Subword-level and byte-level language models** Subword-based models are widely used in NLP tasks, where subwords are typically the smallest units of input. These models take a sequence of subwords based on subword tokenization methods such as BPE, WordPiece, and SentencePiece (Sennrich et al., 2015; Wu et al., 2016; Kudo & Richardson, 2018). Despite their great performance, subword-based models still have some limitations. For instance, they cannot handle out-of-vocabulary subwords and require language-specific tokenizers for languages such as Chinese, Korean, and Japanese. Another challenge is the high space complexity of the large size of the embedding matrix when the vocabulary size is very large.

One possible solution to address these issues is to use byte tokens as inputs and outputs, as demonstrated in many recent works (Shaham & Levy, 2020; Zhang & Xu, 2022; Xue et al., 2022). UTF-8 is a universal standard for almost all language writing systems, making it possible to represent all languages using a fixed and shared byte vocabulary. Therefore, there will be no out-vocabulary words and the language-specific tokenizer is unnecessary. In addition, as the total number of bytes in UTF-8 is 256, the embedding matrix for byte vocabulary is much smaller than most subword vocabularies, reducing the number of parameters in the embedding layer and saving memory space.

**Subword-level model with character- or byte-level fusion** The naive character-based and byte-based models often result in longer input sequences compared to the subword-based model, which increases the time complexity. To make the model efficient, several recent papers have explored character-level or byte-level fusion. For example, (Tay et al., 2021) propose CHARFORMER, using a soft gradient-based subword tokenization module to obtain "subword tokens". It generates and scores multiple subword blocks, aggregates them to obtain subword representation, and then performs downsampling to reduce the sequence length. Although CHARFORMER is faster than vanilla byte-based or character-based models, it does not maintain the subword boundaries, which can limit the interpretability ability of the model. Sreedhar et al. (2022) propose Local Bytes Fusion (LOBEF) to aggregate local semantic information and maintain the word boundary. However, it does not reduce the sequence length, making training and inference time-consuming.

## 3    PRELIMINARIES

### 3.1    SUBWORD-LEVEL AND BYTE-LEVEL TOKENIZATION

Tokenization is an essential process in NLP. It splits input text into a sequence of tokens, which can be subwords or bytes. The resulting sequence of tokens is fed into various NLP models such as Transformer, convolutional neural network (CNN), and recurrent neural network (RNN) for further processing. The following example shows an input text "comedy film" and its subword token sequence with BPE and byte token sequence with UTF-8.

- Input text: "comedy film" with 11 characters.
- Subword tokenization with BPE: "com", "##edy", "film".
- Byte tokenization with UTF-8: 99, 111, 109, 101, 100, 121, 32, 102, 105, 108, 109 (11 bytes).

### 3.2    FEDERATED LEARNING

In federated learning (FL), multiple clients jointly train a model using their private data but without sharing the data. Assume we have $N$ clients, $\mathcal{C} = \{c_1, c_2, \ldots, c_N\}$, and a server $s$, in an FL system. The jointly trained model is $f$ with parameters $\theta$. The clients' private data are $\mathcal{D}_1, \mathcal{D}_2, \ldots, \mathcal{D}_N$ and the objective function is $\mathcal{L}$. To make it easier to illustrate the computation and communication operations of FL, We assume all the clients participate in each communication and clients use FedSGD (McMahan et al., 2017) to update the model parameters.

In each communication round $t$, server $s$ first sends the model parameters $\theta^t$ to all clients. Then each client $c_i$ compute $\Delta_i^t = \nabla_{\theta^t} \mathcal{L}_{\theta^t}(\mathcal{B}_i)$, the gradients of current model $f_{\theta^t}$, based on a randomly

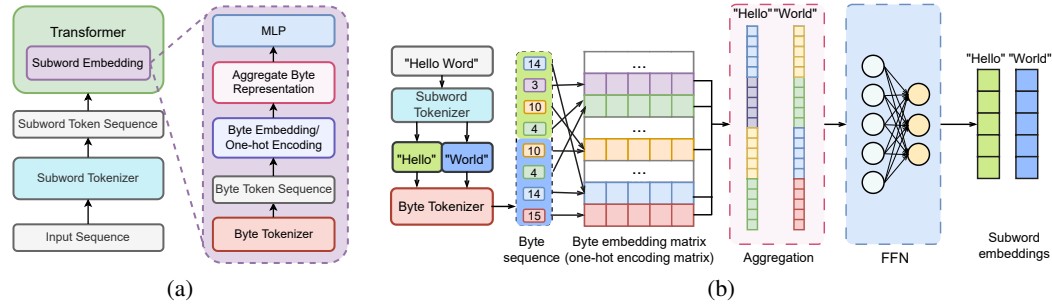

Figure 2: (a): An overview of the transformer model with SEB. (b): An example of calculating subword embeddings with byte embedding.

sampled data batch $\mathcal{B}_i \subset \mathcal{D}_i$ and $| \mathcal{B}_i |$. After local computation, the clients send the gradients $\Delta_1^t, \Delta_2^t, \ldots, \Delta_N^t$ to server and server $s$ aggregate all the gradients and update the model:

$$\theta^{t+1} = \theta^t - \eta \sum_{i=1}^{N} \Delta_i^t = \theta^t - \eta \sum_{i=1}^{N} \nabla_{\theta^t} \mathcal{L}_{\theta^t}(\mathcal{B}_i). \tag{1}$$

Here, Equation (1) is the gradient descent, and $\eta$ is the learning rate.

### 3.3 THREAT MODEL

**Adversary's capabilities and objective** In this paper, we follow the attack settings in Gupta et al. (2022) under client-server architecture. The optimized model is a language model $\mathcal{L}$, parameterized by $\theta$. This scenario makes the attacker white box access to the gradients $\nabla_{\theta^t} \mathcal{L}_{\theta^t}(\mathcal{B}_i)$ sent by the victim client $c_i$. $\theta^t$ is the model parameter that the server sends to the clients at any communication round $t$. From parameters $\theta^t$ and gradients $\nabla_{\theta^t} \mathcal{L}_{\theta^t}(\mathcal{B}_i)$, the attacker can get the information of the vocabulary $\mathcal{V}$ and the embedding matrix $\mathbf{W}$ to retrieve which tokens are updated. The goal of the attacker is to recover at least one sentence from $\mathcal{B}_i$, based on $\nabla_{\theta^t} \mathcal{L}_{\theta^t}(\mathcal{B}_i)$ and $\theta^t$.

**Attack model** This paper does not address the issue of gradient leakage attacks, which aim to obtain private data by minimizing the difference between gradients derived from a dummy input and the actual gradients of the victim's data. This is because several methods have already been proposed to mitigate this particular attack (Zhu et al., 2019; Deng et al., 2021; Wei et al., 2020). Instead, our focus is on a specific attack model introduced in Gupta et al. (2022), for which effective defenses have yet to be explored. In this model, the attacker attempts to reconstruct sentences from the victim's training batches through a three-step process: (1) extracting candidate tokens from the gradients, (2) applying beam search with a pre-trained Language Model, such as GPT-2, to reconstruct the input sentence, and (3) reordering the subword tokens to achieve the best possible reconstruction.

## 4 PROPOSED METHOD

Our goal is to develop a subword embedding approach that requires a smaller byte embedding matrix while maintaining subword boundaries. A smaller byte embedding matrix can save space and potentially protect against attacks based on the embedding gradients in federated learning. Preserving subword boundaries and keeping the subword sequence maintains the model's time efficiency. This raises two main challenges: 1) how to convert subwords into a byte sequence? and 2) how to obtain subword embeddings using byte representations?

In this section, we describe our proposed Subword Embedding from Bytes (SEB) method. Figure 2 shows an overview of SEB, including byte sequence for input text, byte embeddings, aggregation of byte embeddings, and a feed-forward network to output the subword embedding. We will introduce the details of each part and analyze the complexity in the following sections.

### 4.1 CONSTRUCTION OF SUBWORD TO BYTE SEQUENCE MAPPING

UTF-8 encoding results in different sequence lengths for subwords. In real practice, all byte sequences need to be padded to the same length, making the byte sequence of the subword even longer. Instead of using the existing byte encoding system, we define our subword to byte sequence mapping $\mathcal{M} : \mathcal{V}_w \to (\mathcal{V}_b)^n$. $\mathcal{V}_w$ and $\mathcal{V}_b$ are subword and byte vocabularies with size of $V_w$ and $V_b$, respectively. $(\mathcal{V}_b)^n$ is a sequence of $n$ bytes in $\mathcal{V}_b$. Here the byte vocabulary size $V_b$ and the number of bytes $n$ to represent a subword are hyperparameters. In this way, every subword is represented with the same length, getting rid of the longer byte sequence with padding.

To construct the mapping, for every subword $w_i \in \mathcal{V}_w$, we randomly sample $n$ bytes with replacement from $\mathcal{V}_b$ to obtain the byte sequence $(b_{i1}, b_{i2}, \ldots, b_{in})$. If the byte sequence already exists in $\mathcal{M}$, we repeat the sampling until a unique byte sequence is obtained. For example, we set $V_b = 64$ and $n = 4$. A subword "Hello" can be represented with $(14, 3, 10, 4)$, shown in Figure 2(b).

We analyze the probability $p$ that two subwords are mapped to the same byte sequence. With the byte vocabulary size $V_b$ and the number of bytes per subword $n$, the probability $p = 1/(V_b)^n$. For example, if $V_b = 16$ and $n = 4$ then $p = 1.5 \times 10^{-5}$. For $V_b = 128$ and $n = 8$ in our experiment, $p = 1.39 \times 10^{-17}$, which means there is almost no possibility to map two words into the same subword sequence. Therefore, SEB is highly expressive for representing subwords.

### 4.2 SUBWORD EMBEDDING BASED ON BYTE REPRESENTATION

Raw byte sequences as input result in longer lengths, leading to higher space and time complexity (Dai et al., 2019; Sukhbaatar et al., 2019). Different from these models, our method tokenizes the text into a sequence of bytes while preserving the subword boundary. We first tokenize the original text into subwords using a common subword tokenization method, such as BPE. Then, we token each subword into a byte sequence with the mapping we designed above and aggregate byte representations back to subword embeddings. The two detailed algorithms are in Appendix, Algorithm 1 and 2.

Assume the byte embedding matrix is $\mathbf{B} \in \mathbb{R}^{V_b \times d}$, where $d$ is the embedding size. Give an input text $S$, we first tokenize $S$ into a subword sequence $(w_1, w_2, \ldots, w_m)$. Then we further use the mapping $\mathcal{M}$ defined above to tokenize this sequence into a byte sequence $(b_{11}, \ldots, b_{1n}, \ldots, b_{m1}, \ldots, b_{mn})$ with $mn$ bytes. We retrieve the byte embeddings $\mathbf{E} \in \mathbb{R}^{mn \times d}$ for these bytes from $\mathbf{B}$.

To get a subword embedding, adding the byte representations for every $n$ bytes in $\mathbf{E}$ is a simple way. However, this approach does not consider the position of each byte within the subword. Inspired by the idea that incorporating positional information can improve model performance for subword tokens, we induce positional information for byte sequences of subwords by concatenation. This enables the model to capture the position of each byte within the subword and obtain a more accurate and informative representation of the subword. Given the retrieved byte embeddings $\mathbf{E} \in \mathbb{R}^{mn \times d}$, we reshape $\mathbf{E}$ to $\tilde{\mathbf{E}} \in \mathbb{R}^{m \times nd}$ in a row-major order, which is equivalent to concatenation. Then, an FFN is applied to project $\tilde{\mathbf{E}}$ into the dimension $d'$ of the original subword embedding for a specific language model: $\mathbf{E}' = \text{FFN}(\tilde{\mathbf{E}}) \in \mathbb{R}^{m \times d'}$. Note that, the byte embedding matrix $\mathbf{B}$ can be either a real-valued or one-hot embedding matrix because the vocabulary size is small for bytes. We compare the performances for both embedding methods in experiments.

### 4.3 COMPLEXITY ANLYSIS

To demonstrate the efficiency of the proposed SEB, we summarize the space and time complexity of each embedding method in Table 1. Here, the column "Memory" represent the memory usage for each embedding, and the column "Time" shows the time complexity in Transformer attention. For simplicity, we let $d' = d$ and use one linear layer as FFN in SEB which contains $nd^2$ parameters.

Table 1: Complexity for conventional subword embeddings, byte embedding, and our proposed SEB.

| Embedding | Memory | Time |
|---|---|---|
| Subword | $\mathcal{O}(V_w d)$ | $\mathcal{O}(m^2 d)$ |
| Byte | $\mathcal{O}(V_b d)$ | $\mathcal{O}(c^2 m^2 d)$ |
| SEB (Ours) | $\mathcal{O}((nd + V_b)d)$ | $\mathcal{O}(m^2 d)$ |

In terms of space complexity, subword embeddings typically have an exponentially large vocabulary size $V_w$, exceeding $10^4$, while byte embeddings typically have a dictionary size of no more than 256. For the proposed SEB, the number of parameters

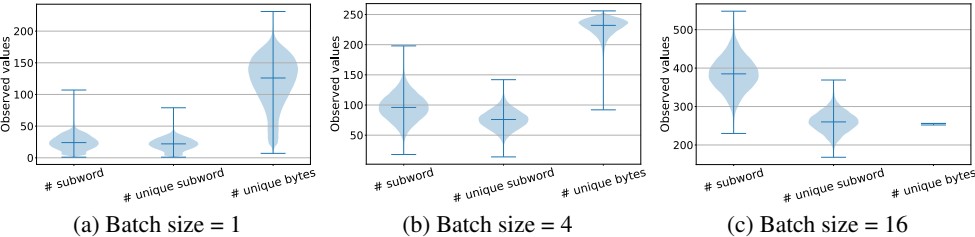

| (a) Batch size = 1 | (b) Batch size = 4 | (c) Batch size = 16 |

Figure 3: The distribution of subword number, unique subword number, and unique byte number in a batch when batch size is 1, 4, 16. The vocabulary sizes of subwords and bytes are 50K and 256.

in embedding is $\mathcal{O}(nd^2 + V_b d) = \mathcal{O}((nd + V_b)d)$, including the FFN and byte embedding matrix. In practice, $nd + V_b \ll V_w$. As a result, both byte embeddings and our proposed SEB significantly reduce the memory cost required for embeddings. In B.5, we show the analysis for space complexity in our experiments. Regarding time complexity, we analyze the attention in the widely used Transformer Vaswani et al. (2017). Given the sequence length $m$, byte embedding is more time-consuming since the input length is $c$ times longer than subword embedding. Here $c$ is the average ratio between the lengths of byte and subword sequences. Based on the statistics Shaham & Levy (2020), $c$ is usually around 5. However, our proposed SEB maintains the same time efficiency as conventional subword embeddings because we preserve the subword sequence along with its boundaries.

## 5 EXPERIMENT

We conduct experiments to demonstrate the advantages of SEB in reducing space complexity, maintaining time efficiency, and preserving privacy for NLP models in federated learning. In all experiments, we set $V_b = 256$ and $n = 8$, which is sufficient to prevent encoding two subwords into the same byte sequences. We use a 2-layer FFN in the proposed SEB.

### 5.1 EXPERIMENTS ON PRIVACY PROTECTION

**Dataset, attack task, and evaluation metrics**  We followed the settings in the FILM attack (Gupta et al., 2022). The dataset is WikiText-103 Merity et al. (2016). For the attack task, we use GPT-2 base Radford et al. (2019) with 117M parameters to recover the input batches. The ROUGE-1/2/L F-Scores Lin (2004) are used to evaluate the similarity between the recovered and original text.

**Quantitative analysis of defense**  We first show that it is difficult to retrieve a bag of candidate subwords in SEB with Figure 3 and 4. In Figure 3, we present the distributions of the subword number, unique subword number, and unique byte number in a client's batch of data. We observe that even a single sample contains over 120 unique bytes on average, while only having approximately 25 unique subwords. In Figure 4, we present the average coverage of subwords for a subset of bytes. Based on Figure 4, 120 bytes cover about 50K subwords. It means recovery is a random generation using almost the entire vocabulary.

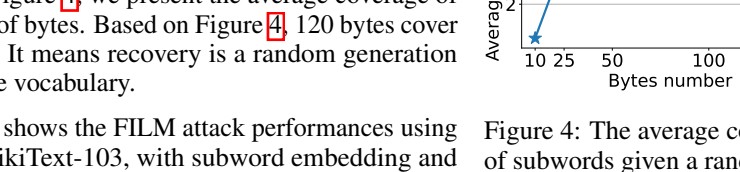

Figure 4: The average coverage of subwords given a random set of bytes with GPT-2 tokenizer.

Additionally, Figure 5 shows the FILM attack performances using various batches on WikiText-103, with subword embedding and SEB. As the candidate subwords are almost the whole vocabulary, beam search takes huge memory which is not executable on our device. To show the defense performance, we loose the constraints and randomly sample 7,000 subwords, combined with the subwords in the original text. We randomly select 5 tested batches for each batch size and take the average ROUGE F-Scores. When batch size is 1, ROUGE-1/2/L scores are close to 1 for attacks with subword embedding, indicating a nearly perfect recovery. However, these scores are quite low when using SEB, showing the effectiveness of SEB to defend the attacks based on embedding gradients.

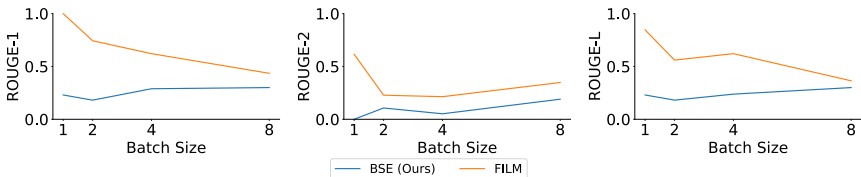

Figure 5: Recovery performance for batch size 1, 2, 4, 8 on WikiText-103.

Table 2: The best recovered sentences by FILM using subword embedding and BSE with batch size 1. Text in green are successfully recovered phrases and words.

|  | Original Sentence | Best Recovered Sentence |
|---|---|---|
| Subword | The historic rainfall caused several dams to fill throughout northeast Mexico. | The rainfall caused several historic dams to fill throughout northeast Mexico. |
| SEB | Pujols is a highly regarded hitter who has shown a "combination of contact hitting ability, patience, and raw power" | He is a professional who has a very high degree of ability, and always takes great advice, without ever assuming power" ("Pivotal Decision Making With Your Head". Retrieved 12 Dec 2007 16 Mar) |

**Qualitative analysis of defense**    To intuitively show the difference between the recovered sentences of FILM using subword embedding and the proposed SEB, we select the best-recovered sentences of these two methods based on the ROUGE-L F-score and list the results in Table 2. In the recovered sentence with the subword embedding, all words are successfully retrieved and have a very close order to the original sentence. However, with SEB, only a few words are retrieved, and many of them are stop words. The results show that SEB can prevent the attacker from recovering private information in the original sentence even though the batch only contains one sentence.

## 5.2    EXPERIMENT ON PERFORMANCE

To provide a more comprehensive assessment of our proposed technique's applicability and performance across different NLP tasks, we conduct experiments on machine translation, sentiment analysis, and Language Modeling (LM) tasks. The enviormental settings are described in B.1.

### 5.2.1    TRANSLATION

**Dataset and evaluation metrics**    In the translation task, we consider two datasets, one is the medium-size IWSLT14 (Cettolo et al., 2014) dataset and a large scale dataset WMT14 Bojar et al. (2014). We follow the settings as prior work (Shaham & Levy, 2020; Zhang & Xu, 2022) and translate German (de) to English (en) in IWSLT14 (Cettolo et al., 2014). The translation of WMT is English (en) to German (de) and the preprocessing is the same as Fairseq (Ott et al., 2019). We use SacreBLEU, case-sensitive, with the 13a tokenizer Post (2018) as the evaluation metric. A detailed description of preprocessing, model architecture, and hyperparameter settings can be found in Appendix B.2.

**Main results**    For IWSLT14, we run 5 trials and report the average performance with the standard deviation. We show the translation results of Transformer with subword embedding and SEB in Table 3. The hidden dimension of the two-layer FFN is 2048 for IWSLT because we try to keep the total parameters of $SEB_{co}$ the same as the original Transformer. For WMT, the hidden dimension of FFN is 4096. Here, we test three variants of SEB when aggregating the byte embedding back to subword embedding: added real-valued embedding ($SEB_{ar}$), concatenated real-valued embedding ($SEB_{cr}$), and concatenated one-hot embedding ($SEB_{co}$). In this experiment, the dimensions of real-valued and one-hot vectors are 512 and 256. Table 3 shows that $SEB_{cr}$ and $SEB_{co}$ can achieve better performances than subword embedding. Concatenating the one-hot vectors yields better results even with fewer model parameters than concatenating byte embedding. Therefore, we can conclude

Table 3: BLEU score of IWSLT14 and WMT14. $SEB_{ar}$ and $SEB_{cr}$: SEB with added and concatenated real-valued embeddings, respectively. $SEB_{co}$: SEB with concatenated one-hot byte embeddings.

| Datasets | Embeddings | # Params | BLEU |
|---|---|---|---|
| IWSLT14 | Subword | 5.2M | $34.54 \pm 0.10$ |
| | $SEB_{ar}$ | 4.3M | $34.64 \pm 0.15$ |
| | $SEB_{cr}$ | 9.6M | $35.32 \pm 0.15$ |
| | $SEB_{co}$ | 5.2M | $\mathbf{35.44 \pm 0.10}$ |
| WMT14 | Subword | 22.3M | 26.0 |
| | $SEB_{co}$ | 6.3M | 26.0 |

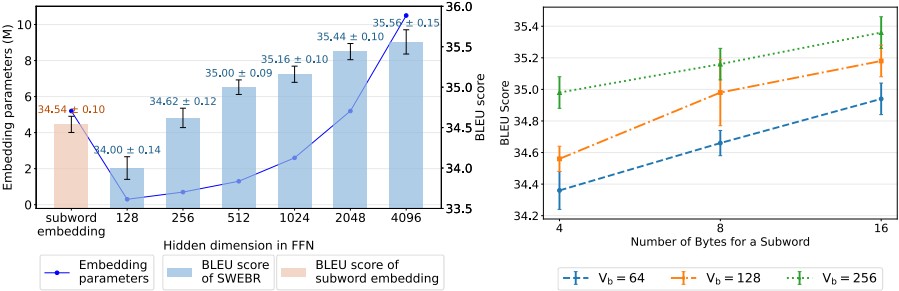

Figure 6: Results on embedding parameters, vocabulary size, and number of bytes per subword. Left: The BLEU scores versus hidden dimension in FFN and embedding parameters. Right: Comparison of mean BLEU scores for different byte vocabulary sizes and different numbers of bytes per subword.

that SEB is a better alternative to using large subword embeddings. Additionally, based on the comparison between $SEB_{ar}$ and $SEB_{cr}$, we find that concatenation is better than the simple adding of byte embeddings. This is expected as Section 4.2 because adding does not consider the positional information of bytes. The result of WMT14 shows the same performance as the subword-based model but with a smaller size of embedding parameters. It is important to emphasize that while privacy is improved, our model achieves the same or better accuracy than the baseline methods.

**Sensitivity analysis on FFN hidden units** In this experiment, we test the sensitivity of $SEB_{co}$ on FFN hidden units, because it is one of the major factors for embedding parameters. Here, we set different FFN hidden units as $\{128, 256, 512, 1024, 2048, 4096\}$, with the total embedding parameter numbers of 0.3M, 0.7M, 1.3M, 2.7M, 5.2M, and 10.5M, respectively. The number of embedding parameters and translation BLEU scores are shown in the left of Figure 6. When the numbers of hidden units are 256, 512, and 1024, $SEB_{co}$ can obtain better performance with fewer parameters. Although the model can still achieve better performance when hidden units are larger than 2048, it does not have advantages over the original transformer on model size.

**Sensitivity analysis on $V_b$ and $n$** To investigate the impact of the byte vocabulary size $V_b$ and number of bytes per subword $n$, we set $V_b$ as $64, 128, 256$ and $n$ as $4, 8, 16$. Based on the previous experiment, we set the hidden units in the 2-layer FFN to 1024 in $SEB_{co}$, which provides good performance with a small scale of parameters. We first report the model size in terms of embedding parameter numbers in Table 4. All of the settings have smaller embedding parameter numbers than the original Transformer. We further demonstrate the translation performance with these

Table 4: Embedding parameter number for different $V_b$ and $n$.

| Byte Tokens per Subword | Byte Vocabulary Size | | |
|---|---|---|---|
| | 64 | 128 | 256 |
| 4 | 0.79M | 1.05M | 1.57M |
| 8 | 1.05M | 1.57M | 2.62M |
| 16 | 1.57M | 2.62M | 4.72M |

settings in Figure 6 (right). It indicates that increasing $n$ leads to better model performance for a fixed $V_b$. This is because increasing $n$ results in more possible positions per byte token, which provide more information in the aggregated vector. Similarly, when we fix $n$ and increase $V_b$, the increased byte

vocabulary diversity makes the aggregated vector more expressive. Therefore, increasing the byte vocabulary size and the number of byte tokens per subword can improve the model expressiveness leading to improved performance. Furthermore, Figure 6 (right) and Table 4 show that models with similar amounts of parameters have similar performance, even with different $V_b$ and $n$. In conclusion, as long as $V_b$ and $n$ ensure that $SEB_{co}$ has sufficient expressive ability, the model performance is more closely related to the number of parameters than to specific $V_b$ and $n$.

### 5.2.2 SENTIMENT ANALYSIS

**Dataset and evaluation metrics**   We use IMDb Maas et al. (2011) and SST2 Socher et al. (2013) datasets provided by Hugging Face. The detailed preprocessing of the dataset is shown in Appendix B.2. We use the accuracy for evaluation which is a routine in prior work Minaee et al. (2019); Yenter & Verma (2017). The implementation details are in Appendix B.3.

**Main results**   We compare the same BiLSTM models with subword embedding and $SEB_{co}$. The classification accuracies are shown in Table 5. The results show that $SEB_{co}$ can replace the conventional subword embedding without hurting the model performance. For SST2, $SEB_{co}$ even has better performance. The reason for that is the parameters of the conventional subword embedding layer in BiLSTM take a large portion of the model parameters,

Table 5: Results on Sentiment analysis.

|  | IMDb (%) | SST2 (%) |
|---|---|---|
| Subword | $85.6 \pm 0.5$ | $81.2 \pm 0.7$ |
| $SEB_{co}$ | $\mathbf{85.8 \pm 0.2}$ | $\mathbf{82.5 \pm 0.7}$ |

making the model easily overfitting. In this experiment, $SEB_{co}$ has smaller embedding parameters, which can address overfitting. We show that SEB also learns the semantic meaning of subword in B.4.

### 5.2.3 LANGUAGE MODELING

**Dataset and evaluation metrics**   We use the same data as Fairseq did for the language modeling tasks. The dataset we use is WikiText-103. We use the same preprocessing and training settings as the official Fairseq does. The number of samples for training, testing, and validation are 1801350, 3760, and 4358 respectively. We evaluate the language modeling performance with perplexity.

Table 6: Perplexity of language modeling for subword embedding and SEB.

|  | # Paramters | Perplexity |
|---|---|---|
| Subword | 13.7M | 30.84 |
| $SEB_{co}$ | 10.5M | 30.55 |

**Main results**   For language modeling (LM), our proposed method achieved better performance on perplexity while using a smaller size of parameters. The results are shown in Table 6, which demonstrate that our method SEBis an effective and efficient alternative to the traditional subword embedding.

## 6   CONCLUSION

**Summary**   This paper introduces SEB, Subword Embedding of Bytes, a novel subword embedding method that defends against privacy leakage attacks based on embedding gradients in federated learning. Unlike traditional approaches that learn a large subword embedding matrix, SEB uses smaller byte embeddings or byte one-hot encoding and aggregates byte representations to obtain subword embeddings. With SEB, attackers cannot retrieve a small set of subwords and generate private text, even with a well-trained large language model. Our extensive experiments show that SEB is effective for machine translation without sacrificing performance or efficiency. Additionally, we demonstrate that SEB makes it difficult for attackers to recover private text with embedding gradients in federated learning.

**Limitations and future work**   Limited to the computation resources, we only experiment with moderate datasets and two tasks, machine translation, and sentiment analysis. The efficiency and effectiveness of our proposed method on large language models as well as other natural language processing tasks still need exploration. What's more, the generalization ability of SEB is not discussed in this work. Therefore, in the future, we will keep working on these aspects.

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
