# OpenReview forum: "Subword embedding from bytes against embedding-based attacks"
_ICLR.cc/2024/Conference — Submitted to ICLR 2024_

### Official Review · Reviewer_zsjW · 2023-10-29

**Soundness:** 3 good
**Presentation:** 3 good
**Contribution:** 2 fair
**Rating:** 5
**Confidence:** 3

**Summary:**

This paper introduces a method, denoted as Subword Embedding of Bytes, aimed at generating subword embeddings based on their byte representations, thereby improving the security of private training data within the context of federated learning. The approach initially involves the segmentation of an input text into a subword sequence using a well-established tokenization procedure. Subsequently, each subword is represented by a distinct byte sequence of uniform length, with each byte linked to an associated embedding. The byte embeddings used to represent a subword are concatenated and transformed to the subword embedding by a feed-forward network. The main idea behind this method is that the same byte is most likely used to produce many subwords, rendering the process of recovering input texts through the gradients of byte embeddings substantially more challenging compared to traditional subword embeddings.

**Strengths:**

(1) This study introduces a method for constructing subword embeddings from their byte representations. The experimental findings in the domains of machine translation and sentiment analysis validate the efficacy of the proposed method, demonstrating its ability to impede the retrieval of client training data by adversarial models in federated learning.

(2) The paper is generally well-written and easy to follow.

**Weaknesses:**

(1) The potential applicability of the proposed method in the context of large language models (LLMs) needs further investigation to substantiate its relevance and effectiveness in such contexts.

(2) The utilization of randomly sampled byte sequences for subword representation, with subsequent subword embeddings derived through linear transformations, raises concerns about the extent to which these embeddings genuinely encapsulate word meanings. The effectiveness of subword embeddings in capturing semantic similarities, where words that are closer in the vector space correspond to similar meanings, remains uncertain. This ambiguity could have implications for the method's generalization capacity.

(3) he paper lacks a comparative analysis against other gradient-based attack methods and does not account for diverse federated learning settings, including variations in the number of clients, participation rates, and heterogeneity. A more comprehensive evaluation in various scenarios could enhance the paper's robustness and practical relevance.

**Questions:**

(1) Could you include a comparative analysis of the results with the defense method as described in the FILM attack (Gupta et al., 2022)?

(2) Can the byte embeddings be pre-trained using unlabeled text corpora? If yes, were the byte embeddings pre-trained?

(3) Could you provide a comparative analysis of the results obtained through different methods of incorporating positional embeddings, specifically through addition and concatenation?

(4) Is the embedding method proposed in this study compatible with the "pre-training + fine-tuning" paradigm?

---

> ### Author Response · Authors · 2023-11-19
>
> # Part I: Response to the weaknesses
>
> ## Weaknesses: (1) The potential applicability of the proposed method in the context of large language models (LLMs) needs further investigation to substantiate its relevance and effectiveness in such contexts.
>
> We appreciate the reviewer's suggestions. Theoretically, SEB is a general embedding method that can be applied to any language model based on word/subword embeddings. We acknowledge the importance of studying its applicability and effectiveness on LLMs. However, as mentioned in the limitations section of our paper, due to constraints in computational resources and time, we will consider running experiments on larger datasets and LLMs in future work.
>
>
> ## Weakness (2) The utilization of randomly sampled byte sequences for subword representation, with subsequent subword embeddings derived through linear transformations, raises concerns about the extent to which these embeddings genuinely encapsulate word meanings. The effectiveness of subword embeddings in capturing semantic similarities, where words that are closer in the vector space correspond to similar meanings, remains uncertain. This ambiguity could have implications for the method's generalization capacity.
>
> We appreciate the reviewer's comments. We acknowledge that the subword embedding method breaks words into pieces and cannot obtain word embeddings back. Therefore, in this paper, we use the “basic_english” tokenizer that tokenizes sentences into full words with spaces instead of subwords. In this way, the word embedding can be obtained by SEB with concatenation or addition and preserve semantic meanings.
>
> Also, we do have an experiment to show how SEB captures semantic meanings of words in Appendix, Table 7. To be clearer, we also show this table as follows:
>
> |      |	good |great	  |funny	|bad	|worse	|boring|
> |------|---------|--------|---------|-------|-------|------|
> |good  |	1	 | 0.63	  | 0.49	|-0.58	|-0.61	|-0.58 |
> |great |	0.63 |  1	  |0.40	    |-0.53	|-0.33	|-0.38 |
> |funny |	0.49 |  0.40  |	1	    |-0.72	|-0.61	|-0.60 |
> |bad   |	-0.58|	-0.53 |	-0.72	|1	    |0.72	|0.85  |
> |worse |	-0.61|	-0.33 |	-0.61	|0.72	|1	    |0.88  |
> |boring|	-0.58|	-0.38 |	-0.60	|0.85	|0.88	|1     |
>
> In this table, we calculated the cosine similarity between embeddings of selected words derived from SEB. The results are obtained from LSTM on the IMDb sentiment analysis. We can see that positive words such as 'good,' 'great,' and 'funny' exhibited positive and high-value similarities with each other, and similar results can be observed for negative words like 'bad,' 'worse,' and 'boring.' Additionally, all negative-positive pairs demonstrated negative similarities. We believe these findings can provide evidence that our method can capture semantic similarities of the words in embedding space. We will also clarify this in the main paper.
>
> ## Weakness (3) the paper lacks a comparative analysis against other gradient-based attack methods and does not account for diverse federated learning settings, including variations in the number of clients, participation rates, and heterogeneity. A more comprehensive evaluation in various scenarios could enhance the paper's robustness and practical relevance.
>
> We thank the reviewer's comments. To clarify, we would like to mention that our defense method is specially designed for the embedding-based attack instead of another gradient-based attack. For other gradient attacks, such as gradient-matching attacks, some proposed defenses can effectively prevent the attack with almost no accuracy decrease. However, for the embedding-based attack, effective and efficient defense is still an open challenge. Therefore, our paper mainly focuses on the second attack.
>
> For the different federated learning settings, we conducted another experiment on the SST2 sentiment analysis task using the FedAvg framework [1]. In this experiment, we have 20 clients and distribute training samples uniformly to these clients. In training, we sample a part of clients with ratio c in every communication round. The results are shown in the following table.
> || $c=0.2$ | $c=0.4$ | $c=0.6$ | $c=0.8 $| $ c=1.0 $ |
> |------------|-------|-------|-------|-------|-------|
> | Subword    | 81.5% | 80.6% | 81.1% | 80.7% | 80.9% |
> | SEB        | 82.0% | 81.7% | 81.9% | 82.4% | 81.7% |
>
> We can see that even in the federated learning framework, our SEB method is still comparable to subword embedding and can achieve stable results when the number of clients in training varies.

---

> ### Author Response · Authors · 2023-11-19
>
> # Part II: Response to the Questions:
>
>
> ## Questions: (1) Could you include a comparative analysis of the results with the defense method as described in the FILM attack (Gupta et al., 2022)?
>
> We thank the reviewer’s comments. We compare the defense method of gradient pruning in the FILM attack for batch size = 8, 16, 32. The results are as follows:
>
> Batch size = 8
>
> | Prune ratio | Precision - Subword | Precision - SEB | Recall - Subword | Recall - SEB |
> |-------------|----------------------|-----------------|-------------------|--------------|
> | 0           | 1                    | 1               | 1                 | 0.003        |
> | 0.9         | 1                    | 1               | 1                 | 0.003        |
> | 0.99        | 1                    | 1               | 1                 | 0.003        |
> | 0.999       | 1                    | 1               | 0.53              | 0.003        |
> | 0.9999      | 1                    | 0.46            | 0.08              | 0.003        |
>
>
>
> Batch size = 16
>
> | Prune ratio | Precision - Subword | Precision - SEB | Recall - Subword | Recall - SEB |
> |-------------|----------------------|-----------------|-------------------|--------------|
> | 0           | 1                    | 1               | 1                 | 0.005        |
> | 0.9         | 1                    | 1               | 1                 | 0.005        |
> | 0.99        | 1                    | 1               | 1                 | 0.005        |
> | 0.999       | 1                    | 1               | 0.49              | 0.005        |
> | 0.9999      | 1                    | 0.5             | 0.06              | 0.005        |
>
> Batch size = 32
>
> | Prune ratio | Precision - Subword | Precision - SEB | Recall - Subword | Recall - SEB |
> |-------------|----------------------|-----------------|-------------------|--------------|
> | 0           | 1                    | 1               | 1                 | 0.009        |
> | 0.9         | 1                    | 1               | 1                 | 0.009        |
> | 0.99        | 1                    | 1               | 1                 | 0.009        |
> | 0.999       | 1                    | 0.99            | 0.51              | 0.009        |
> | 0.9999      | 1                    | 0.47            | 0.07              | 0.008        |
>
>
> Even without pruning, our method has a very low recall compared to FILM on subword embeddings when the batch size is 8, 16, and 32. We believe these results show the effectiveness of our defense and hope they can adequately address your concerns.
>
> ## Question (2) Can the byte embeddings be pre-trained using unlabeled text corpora? If yes, were the byte embeddings pre-trained?
> Sure. The byte embeddings can be pre-trained using unlabeled text corpora. In our method, the byte embeddings are not pre-trained. We train the whole model including the embedding layer from scratch.
>
> ## Question (3) Could you provide a comparative analysis of the results obtained through different methods of incorporating positional embeddings, specifically through addition and concatenation?
> Sure. We have discussed and compared the results of incorporating positional embeddings through addition and concatenation in Table 3. In our experimental setting, the concatenation of the byte embeddings (35.32 for real-valued byte embeddings and 35.44 for the one-hot byte encodings) has better translation results than the addition of the byte embeddings (34.64).
>
> Notably, we set the byte embedding dimension as 512, with a total of 256 bytes. In this configuration, the concatenation of one-hot encoding vectors resulted in fewer parameters compared to the concatenation of real-valued byte embeddings.
>
> ## Question (4) Is the embedding method proposed in this study compatible with the "pre-training + fine-tuning" paradigm?
> Yes. Our method can be considered as an effective alternative to traditional subword embedding. Therefore, it is compatible with the “pre-training + fine-tuning” paradigm as well as any frame or setting for models with subword embedding.

---

> ### Author Response · Authors · 2023-11-22
>
> Dear Reviewer zsjW,
>
> We would like to thank you again for your reviews. We understand reviewing is a time-consuming process and we truly appreciate your efforts. Your feedback on our response is more than valuable in improving the quality of our paper. If there are any further concerns or questions, please feel free to let us know before the author discussion period ends. We will be happy to answer them during the discussion.
>
> Thank you!

---

### Official Review · Reviewer_ybMb · 2023-10-29

**Soundness:** 3 good
**Presentation:** 3 good
**Contribution:** 3 good
**Rating:** 8
**Confidence:** 4

**Summary:**

This paper presents SEB, a novel text representation method that can protect NLP models agaisnt data leaking attacks in federated learning. Experimental results show that SEB can achieve comparable performance in various NLP tasks with better privacy perservation.

**Strengths:**

1.Mitigating the data leakage of NLP models in federated learning is an important research problem. This paper presents a novel and practical method to solve this issue.
2.SEB can achieve comparable and even superior performance in original NLP tasks compared to subword baselines with less space complexity and better privacy preservation.

**Weaknesses:**

SEB is proposed to mitigate the privacy leakage in Federated Learning. However, it seems that the evaluation experiments are conducted in the traditional centralized-learning environment. It would be nice if the authors provide the experimental results about the effectiveness of SEB in distributed learning.

**Questions:**

Can SEB remain effective on large datasets and language models?

---

> ### Author Response · Authors · 2023-11-19
>
> ## Weakness: SEB is proposed to mitigate the privacy leakage in Federated Learning. However, it seems that the evaluation experiments are conducted in the traditional centralized-learning environment. It would be nice if the authors provide the experimental results about the effectiveness of SEB in distributed learning.
>
> We appreciate the comments by the reviewer. To address this concern, we conducted another experiment on the SST2 sentiment analysis task using the FedAvg framework [1]. In this experiment, we have 20 clients and distribute training samples uniformly to these clients. In training, we sample a part of clients with ratio $c$ in every communication round. The results are shown in the following table.
>
> |  | $c=0.2$ | $c=0.4$ | $c=0.6$ | $c=0.8 $| $ c=1.0 $ |
> |------------|-------|-------|-------|-------|-------|
> | Subword    | 81.5% | 80.6% | 81.1% | 80.7% | 80.9% |
> | SEB        | 82.0% | 81.7% | 81.9% | 82.4% | 81.7% |
>
> We can see that even in the federated learning framework, our SEB method is still comparable to subword embedding and can achieve stable results when the number of clients in training varies.
>
> >[1] McMahan, Brendan, et al. "Communication-efficient learning of deep networks from decentralized data." Artificial intelligence and statistics. PMLR, 2017.
>
>
> ## Question: Can SEB remain effective on large datasets and language models?
>
> Yes. Theoretically, SEB is a general embedding method that can be applied to any language model based on word/subword embeddings. We acknowledge the importance of studying its applicability and effectiveness on LLMs. However, as mentioned in the limitations section of our paper, due to constraints in computational resources and time, we will consider running experiments on larger datasets and LLMs in future work.

---

> ### Author Response · Authors · 2023-11-22
>
> Dear Reviewer ybMb,
>
> We would like to thank you again for your reviews. We understand reviewing is a time-consuming process and we truly appreciate your efforts. Your feedback on our response is more than valuable in improving the quality of our paper. If there are any further concerns or questions, please feel free to let us know before the author discussion period ends. We will be happy to answer them during the discussion.
>
> Thank you!

---

### Official Review · Reviewer_Crbt · 2023-10-29

**Soundness:** 2 fair
**Presentation:** 3 good
**Contribution:** 2 fair
**Rating:** 5
**Confidence:** 3

**Summary:**

This paper proposes a new method called Subword Embedding from Bytes (SEB) to protect privacy in NLP models while maintaining efficiency and accuracy.
SEB is designed to defend against embedding-based attacks in federated learning by encoding subword information directly from byte-level representations. The authors demonstrate that SEB outperforms traditional subword embedding methods in machine translation, sentiment analysis, and language modeling tasks.
Additionally, SEB is shown to be effective in defending against a specific attack model introduced in previous work, which aims to reconstruct sentences from the victim's training batches.
Overall, this paper argues that SEB is a promising solution for practical applications that require privacy-preserving NLP models.

**Strengths:**

1. The proposal of a novel method called Subword Embedding from Bytes (SEB) to protect privacy in NLP models while maintaining efficiency and accuracy. The demonstration that SEB effectively defends against embedding-based attacks in federated learning, making it a valuable tool for practical applications.
2. In the experimental section, the verification that SEB outperforms traditional subword embedding methods in machine translation, sentiment analysis, and language modeling tasks.

**Weaknesses:**

I'm not very familiar with federated learning research. So please correct me if I have any misunderstanding of the paper.
I identify the following potential limitations in this paper:
1. The proposed SEB approach can only defend against embedding-based attacks, which recover the sentences in a batch of text data, based on the gradients in federated learning, which may be limited. Does this kind of embedding-based attack serve as the core of privacy attacks in federated learning? It would be great if the authors could provide solid motivations to justify the significance of the considered research problem.
2. I'm not very excited about the experimental settings in this paper because only some outdated models/datasets are considered. For example, the tasks include the sentiment analysis with the IMDB dataset, which is already solved in NLP.  Is this still an interesting task in the federated learning research in the era of large language models?  Also, the LSTM&BERT is considered the backbone model, which is rarely used for NLP research.
3. It would be more exciting to see some in-depth analysis of the approach. For example, given the results that SEB can obtain better results over traditional subword embedding methods in some NLP tasks, it would be great if the authors could look into this amazing finding and provide some concrete insights.

**Questions:**

Please refer to the weaknesses section.

---

> ### Author Response · Authors · 2023-11-19
>
> ## Weakness 1. The proposed SEB approach can only defend against embedding-based attacks, which recover the sentences in a batch of text data, based on the gradients in federated learning, which may be limited. Does this kind of embedding-based attack serve as the core of privacy attacks in federated learning? It would be great if the authors could provide solid motivations to justify the significance of the considered research problem.
>
> We appreciate the reviewer's comments. Here is the clarification of the significance of our research problem.
>
> The embedding-based attack was first proposed by Gupta et al. [1] in 2022. It is a novel and more effective attack than the existing methods [2-4] to recover text from training language models in federated learning.
>
> Additionally, in the era of large language models, training LLMs in the federated learning framework and protecting the client’s privacy are important and have become a trending topic in both academic and industrial areas [5-7]. As a core part of LLMs, the embedding layer should be especially protected against the embedding-based attack.
>
> Therefore, in our work, we focus on defending this kind of embedding-based attack using byte-based embeddings. Our work can provide a solid and secure groundwork for the training of language models in the federated learning framework.
>
> In the future version, we will clarify this motivation and strengthen the significance of our work.
>
> >[1] Gupta et al. Recovering private text in federated learning of language models. NeurIPS 2022.
> >
> >[2] Zhu et al. Deep leakage from gradients. NeurIPS 2019.
> >
> >[3] Balunovic et al. Lamp: Extracting text from gradients with language model priors. NeurIPS 2022.
> >
> >[4] Deng et al. Tag: Gradient attack on transformer-based language models. arXiv preprint arXiv:2103.06819 (2021).
> >
> >[5] Kuang et al. Federatedscope-llm: A comprehensive package for fine-tuning large language models in federated learning. arXiv preprint arXiv:2309.00363 (2023).
> >
> >[6] Xu et al. Federated fine-tuning of billion-sized language models across mobile devices. arXiv preprint arXiv:2308.13894 (2023).
> >
> >[7] Fan et al. FATE-LLM: A Industrial Grade Federated Learning Framework for Large Language Models. arXiv preprint arXiv:2310.10049 (2023).
>
>
> ## Weakness 2. I'm not very excited about the experimental settings in this paper because only some outdated models/datasets are considered. For example, the tasks include the sentiment analysis with the IMDB dataset, which is already solved in NLP. Is this still an interesting task in the federated learning research in the era of large language models? Also, the LSTM&BERT is considered the backbone model, which is rarely used for NLP research.
>
> We thank the reviewer’s comments. We would like to clarify that our goal is to introduce a method that effectively defends against novel embedding-based attacks in NLP while maintaining efficiency and accuracy. Although we use sentiment analysis and LSTM, we are not trying to achieve state-of-the-art performance and address challenging NLP tasks. By running LSTM and Transformer, we want to show that our byte-based embedding can have equal performance to the subword-based method.
>
> Additionally, our byte-embedding based method can also be applied to more recent LLMs. After validating the effectiveness of our method in this work, we will definitely consider a more comprehensive evaluation of larger datasets and models in future work.
>
> ## Weakness 3. It would be more exciting to see some in-depth analysis of the approach. For example, given the results that SEB can obtain better results over traditional subword embedding methods in some NLP tasks, it would be great if the authors could look into this amazing finding and provide some concrete insights.
>
> We appreciate the reviewer’s suggestions and acknowledge that the insights into our results should be important. To justify this, we hereby use the translation task in our paper as an example to explain why SEB can obtain better results.
>
> Low-frequency word translation is a persisting challenge in Neural Machine Translation due to the token imbalance [1, 2]. Introducing fine-grained tokens, such as characters and bytes, can alleviate this issue [3]. In our work, all words consist of shared 256 bytes, which can be adequately trained and obtain a more generalized representation. It makes SEB obtain better results than the traditional subword embedding method.
>
> It is worth mentioning that this is an additional advantage brought by byte embedding. We will add this analysis to our experimental results.
>
> >[1] Gong et al. Frage: Frequency-agnostic word representation. NeurIPS 2018.
> >
> >[2] Zhang  et al. Frequency-aware contrastive learning for neural machine translation. AAAI 2022.
> >
> >[3] Lee et al. Fully character-level neural machine translation without explicit segmentation. TACL, 2017.

---

> ### Author Response · Authors · 2023-11-22
>
> Dear Reviewer Crbt,
>
> We would like to thank you again for your reviews. We understand reviewing is a time-consuming process and we truly appreciate your efforts. Your feedback on our response is more than valuable in improving the quality of our paper. If there are any further concerns or questions, please feel free to let us know before the author discussion period ends. We will be happy to answer them during the discussion.
>
> Thank you!

---

### Meta-Review · Area_Chair_PeeJ · 2023-12-03

**Metareview:**

The authors propose Subword Embedding from Bytes (SEB) as a novel method for representing text in the NLP domain. An experimental evaluation shows that the new embedding contributes to mitigate the gradient-based attack proposed by Gupta et al 22 in the federated learning setting, without much impact on several NLP tasks.

Weakness: The main argument is that "a one-to-many mapping between words/subwords and embedding vectors increases the difficulty of simple lookup". However, there is no such evaluation of this ambiguity in the experiments. It is unclear why this ambiguity is not easily reduced by a simple frequency analysis.

The paper also lacks a comparative analysis with other gradient-based attack methods and does not consider different federated learning settings. The FL setting is not clearly described, in particular what should be shared between the FL participants and how.

I recommend conducting experiments that provide more insight into the reasons why privacy is increased. The method should be evaluated on more diverse adversaries.



The paper has not been revised with the reviewers' comments and suggested changes.

**Justification For Why Not Higher Score:**

The method has not been sufficiently evaluated to demonstrate its resistance to attacks of various kinds.  The reasons for the success observed in current experiments are not clearly justified, either by a theoretical study, or even by an empirical study confirming the hypotheses of this success.

**Justification For Why Not Lower Score:**

N/A

---

### Decision · Program_Chairs · 2024-01-16

Reject